# Three *Seinura* species from Japan with a description of *S. shigaensis* n. sp. (Tylenchomorpha: Aphelenchoididae)

**Natsumi Kanzaki** [1]*, **Taisuke Ekino**[2], **Keiko Hamaguchi**[1], **Yuko Takeuchi-Kaneko**[3]

**1** Kansai Research Center, Forestry and Forest Products Research Institute, Fushimi, Kyoto, Japan,
**2** School of Agriculture, Meiji University, Kawasaki, Kanagawa, Japan, **3** Laboratory of Terrestrial Microbial Ecology, Graduate School of Global Environmental Studies, Kyoto University, Sakyo, Kyoto, Japan

* nkanzaki@affrc.go.jp

## Abstract

A preliminary survey of *Seinura* spp. was conducted in the Kyoto area, Western Japan. The survey yielded four new strains of *Seinura* spp., including two strains of *S. caverna*, a strain of *S. italiensis*, and a strain of an undescribed species. Molecularly, the two strains of *S. caverna* were nearly identical to the type strain but showed some minor variations, particularly in the mitochondrial cytochrome oxidase subunit I gene. The small subunit and D2-D3 large subunit sequences of the Japanese strain of *S. italiensis* were nearly identical and identical to its original description, respectively, and the difference in the small subunit was due to mis-reading of the sequences. The new species, *S. shigaensis* n. sp., was phylogenetically close to *S. caverna* and *S. persica*, although these three species were clearly different phylogenetically. The new species was typologically similar or nearly identical to several other *Seinura* spp., including *S. chertkovi*, *S. christiei*, *S. italiensis*, *S. steineri*, and *S. tenuicaudata*, but it can be distinguished from those species by the morphometric values. Because the new species is phylogenetically very close to *S. caverna*, it could be a good comparative system for *S. caverna* as a potential satellite model for the predatory nematode.

## Introduction

The nematode family Aphelenchoididae is a highly divergent group that includes mycophagous species, plant parasites, insect parasites and predators, and an interesting system for evolutionary studies on feeding mode, including digestion-related enzymes and food recognition [1]. Within the family, only two species, *Bursaphelenchus okinawaensis* Kanzaki, Maehara, Aikawa & Togashi and *Seinura caverna* Kanzaki, Ekino & Masuya, have been experimentally confirmed to be hermaphrodites, i.e., these two species have hermaphroditic and spontaneous males [2, 3]. In addition, these two species have been cultured, and *B. okinawaensis* has been assessed for its genetic background as a satellite model system for fungal-feeding nematodes, including the devastating forest pathogen, *B. xylophilus* (Steiner & Buhrer) Nickle [2, 4].

**Data Availability Statement:** All relevant data are within the manuscript and its Supporting Information files.

**Funding:** This study was supported in part by the Grants-in-Aid for Scientific Research (B), No. 20H03026 (NK) and (C), No. 19K06145 (YT, NK) from the Japan Society for the Promotion of Science. The funder had no role in study design, data collection and analysis, decision to publish, or preparation of the manuscript.

**Competing interests:** The authors have declared that no competing interests exist.

*Seinura caverna* is the only cultured aphelenchoidid predator. This species is expected to be a biological control agent for pest nematode species and a satellite model system for predatory aphelenchoidids because of its wide feeding range, ease of observing feeding behaviour, and the hermaphroditic reproductive mode [3].

However, to establish a useful experimental model system, more *Seinura* Fuchs species must be isolated and cultured as comparative systems for *S. caverna*. Therefore, a preliminary survey of *Seinura* and its related predator species was conducted in Kyoto and Shiga Prefectures, Japan. The survey yielded four strains of *Seinura* spp., including an undescribed species.

The strains were characterised molecularly, and the new species is described and illustrated herein as *S. shigaensis* n. sp.

## Materials and methods

### Ethics statement

Specific permissions were not required for the nematodes collected for the present study. The field used for nematode collection was not privately owned but open to the public and did not involve endangered or protected species.

### Nematode collection, culturing and establishment of isogenic lines

Nematodes were collected from two localities in Makino, Shiga Pref. and Kyoto, Kyoto Pref., Japan during 2018–2020.

Approximately 100 g organic soil containing decomposed plant material was collected from a hollow on the trunk of *Quercus serrata* Murray located in Makino, Shiga, Japan (GPS coordinates: 35˚28'41" N, 136˚01'48" E; altitude: 156 m a.s.l.) on 8 March 2019. The material was brought back to the laboratory, and the nematodes were extracted using a Baermann funnel. Isolated nematodes were examined under a dissecting microscope (S8 Apo, Leica, Jena, Germany) for preliminary identification, and *Seinura* spp. were hand-picked and transferred to NGM agar previously inoculated with *Escherichia coli* strain OP50 and *Acrobeloides* sp., which is used as food for predatory nematodes [3]. The propagated *Seinura* was subcultured in the same medium (*Acrobeloides* on *E. coli* OP50) and maintained as a laboratory culture (laboratory culture code NKZ276).

Materials were collected three times independently from the experimental forest stands of the Kansai Research Center, Forestry and Forest Products Research Institute (Kyoto, Japan).

Two *Seinura* strains were isolated using a Baermann funnel and cultured as described above from partially decomposed *Q. myrsinaefolia* Blume wood chips (<1 cm in the longest axis) (GPS coordinates: 34˚56'28" N, 135˚46'25" E; altitude: 62 m a.s.l.) during March 2018 and rotten wood of an unidentified broad-leaved tree (GPS coordinates: 34˚56'28" N, 135˚46'22" E; altitude: 62 m a.s.l.) during January 2020, as described above. The isolated nematodes were kept as laboratory cultures under the codes NKZ272 (*Q. myrsinaefolia* woodchip) and NKZ278 (unidentified broad-leaved tree).

In addition, nematodes were baited using freeze-killed wax moth, *Galleria mellonella* (L.) larvae. Approximately 500 ml soil was collected at a *Q. variabilis* Blume stand (GPS coordinates: 34˚56'24" N, 135˚46'23" E; altitude: 61 m a.s.l.) during November 2019. The soil was placed in a plastic cup, and a moth larva was buried in the soil and kept at room temperature for 3 days. Thereafter, the larva was recovered from the soil, transferred to water agar (2.0% agar), and the nematodes crawling out from the carcass were examined under a dissecting microscope to hand-pick and establish a culture as described above. The culture was coded NKZ274 and maintained as a laboratory strain.

## Morphotyping for generic identification

Because all *Seinura* spp. are morphologically similar, the cultures were identified at the genus level under a light microscope (Eclipse 80i, Nikon, Tokyo, Japan) equipped with DIC optics and a drawing tube, following the methodology of Kanzaki [5].

## Examination of reproductive mode

Ten juveniles from each strain were transferred individually to their food medium, i.e., *Acrobeloides* sp. propagating on an OP50 lawn prepared in a small ($\phi$ = 40 mm) Petri dish, and propagation was examined 2 weeks after inoculation.

## Molecular profiles and phylogenetic analyses

First, to confirm the identity of each strain, ca 0.7 kb of the D2-D3 expansion segments of the large subunit (D2-D3 LSU) were determined in 10 individuals from each strain. DNA samples were prepared as described by Kikuchi et al. [6] and Tanaka et al. [7]; briefly, the worms were hand-picked from the culture and digested in a nematode lysis solution. The nematode lysate served as the template DNA for polymerase chain reaction analysis. After confirming the species identity within each strain, ca 4.2 kb of the ribosomal RNA region, including the nearly full-length small subunit (SSU), internal transcribed spacer (ITS) including the ITS1, 5.8S and ITS2 regions and the D1-D4 segments of the LSU (D1-D4 LSU), as well as ca 0.7 kb of mitochondrial cytochrome oxidase subunit I (mtCOI) were determined for each strain using the methodology described by Kanzaki and Futai [8] and Ekino et al. [9]. The molecular sequences were deposited in GenBank with accession numbers LC596442 (rDNA of NKZ272 = *S. caverna*), LC596444 (mtCOI of NKZ272 = *S. caverna*), LC596443 (rDNA of NKZ274 = *S. caverna*), LC596445 (mtCOI of NKZ274 = *S. caverna*), LC596440 (rDNA of NKZ276 = *S. shigaensis* n. sp.), LC596441 (rDNA of NKZ278 = *S. italiensis* Gu, Maria, Liu & Pedram) and LC596446 (mtCOI of NKZ278 = *S. italiensis*).

The SSU and D2-D3 LSU sequences were used for the combined phylogenetic analyses. The sequences of these four strains were compared with the corresponding sequences of the other clade 3 aphelenchoidids. The sequences used for comparison with those of the four strains were selected based on the results of a BLAST homology search (https://blast.ncbi.nlm.nih.gov/Blast.cgi?PROGRAM=blastn&PAGE_TYPE=BlastSearch&LINK_LOC=blasthome) and recent publications [3, 10, 11] (S1 Table).

The compared loci were aligned separately using MAFFT [12, 13] (available online at https://mafft.cbrc.jp/alignment/server/index.html) and then combined. The substitution model and parameters were determined using MEGA7 [14]. The phylogenetic relationships were analysed using MrBayes 3.2 [15, 16]; four chains were run for $4 \times 10^6$ generations. Markov chains were sampled at 100-generation intervals [17]. Two independent runs were performed; after confirming the convergence of the runs and discarding the first $2 \times 10^6$ generations as burn-in, the remaining topologies were used to generate a 50% majority-rule consensus tree.

In addition, two loci were analysed separately with the Bayesian analysis using same analytical parameters for combined analysis.

## Morphological observations for species description

Among the four strains, NKZ276 isolated from Makino, Shiga, Japan was considered a new species based on differences in its molecular sequence, and thus, further morphological analyses were conducted in that strain. Live adults from a 2-week-old culture were used for

morphological observations, illustrations and photomicrographs. Nematodes were hand-picked from the culture, and their typological characters were observed and drawn under a light microscope, following the methodology of Kanzaki [5]. Photomicrographs were obtained using the digital microscope camera system MC170 HD (Leica) connected to a microscope. The photomicrographs and hand drawings were edited using PhotoShop Elements 2018 software (Adobe Photosystems, San Jose, CA).

For the type materials, nematodes from the cultures were killed at 60˚C, fixed in TAF (triethanolamine/formalin/distilled water solution = 2:7:91) and processed in glycerine using the modified Seinhorst's method [18]. The nematodes were mounted in glycerine as permanent slides (type materials) according to Maeseneer & d'Herde [19]. The type materials were used to calculate the morphometric values.

## Nomenclatural acts

The electronic vision of this paper meets the requirements of the amended international code of zoological nomenclature (ICZN), and therefore the new name contained herein is available under that code from the electronic vision of this paper. This published work and the contained nomenclatural acts have been registered in the online registration system for the ICZN in ZooBank. The ZooBank LSID (life science identifiers) for this publication is: urn:lsid: zoobank.org:pub: 28B2A3F7-318F-497A-B483-9DE1339D6965. The related LSID information can be viewed through any standard web browser by appending the LSID to the prefix "http://www.zoobank.org/References/".

## Results

### Morphotyping

The four strains of predators were identified as *Seinura* spp. after a morphotyping analysis. However, there were no clear typological differences among the four strains; e.g., the number of lateral lines, the position of the secretory-excretory pore and the length of the post-uterine sac of females/hermaphrodites were not different, although no males were found in the two hermaphroditic strains. Thus, the species was identified based on the molecular profile.

### Reproductive mode

The four strains were separated into two hermaphrodites and two gonochorists based on the reproductive mode, i.e., two strains (NKZ272 and NKZ274) propagated from a single juvenile, whereas the other two (NKZ276 and NKZ278) did not. In addition, female-formed individuals in the hermaphroditic strains produced sperm by themselves.

### Genotyping and sequence variation

The molecular sequences of the barcoding region (SSU and D2-D3 LSU) of the hermaphrodites were identical (NKZ272) or nearly identical (NKZ274) to those of *S. caverna* (LC414971), and thus, these hermaphrodites were tentatively identified as *S. caverna*.

Two *S. caverna* strains and the type strain of the species formed a clade but showed some minor sequence differences. Namely, the SSU sequences of the type strain and NKZ272 were identical and were 1 bp (A/G transition) different from that of NKZ274. The D2-D3 LSU sequences showed a 1 bp difference; i.e., type vs NKZ272 and NKZ272 vs NKZ274 showed a T/C transition, and NKZ272 vs NKZ274 showed a 1 bp gap (insert/deletion). The mtCOI gene sequence showed relatively large variation. The mtCOI sequence of NKZ274 was slightly different from those of the other two, i.e., the type strain and NKZ272 sequences had a 7 bp

transition, the type strain and NKZ274 sequences had 14 transitions and 4 transversions, and the NKZ272 and NKZ274 sequences had 16 transitions and 4 transversions, respectively. These variations are considered intraspecific variations compared with other aphelenchoidids, e.g., *B. conicaudatus* Kanzaki, Tsuda & Futai [20].

The sequence of the gonochorist (NKZ278) was nearly identical to that of *S. italiensis* (SSU: MN428135; D2-D3: MN428136), and the differences were considered to be due to mis-reading of the sequence. Thus, the species was identified as *S. italiensis*. The typological characters and morphometric values of the Japanese strain of *S. italiensis* are provided in S2 Table and S1–S4 Figs. Although the typological characters of the Japanese population fit the original description of *S. italiensis*, morphoametric values vary from each other.

A BLAST homology search for NKZ276 (*S. shigaensis* n. sp.) suggested that this species was close to *S. caverna* (LC414971: 1643/1655 bp of identity without a sequence gap), *S. persica* Adeldoost, Heydari, Miraiez, Jalalinasab & Asghari (MN130058: 1545/1577, 2 gaps), *S. italiensis* (MN428135: 1642/1694, 3 gaps) and *S. hyrcania* Adeldoost, Heydari, Miraiez, Jalalinasab & Asghari (MN130132: 1502/1546, 5 gaps) in terms of the SSU sequence, to *S. caverna* (LC414971: 793/855 with 23 gaps) in terms of ITS, and to *S. caverna* (LC414971: 730/744 without a sequence gap), *S. persica* (MN130141: 708/742 without a sequence gap) and undescribed *Seinura* sp. (KT355496: 625/659 without sequence gap) in terms of the D2-D3 LSU sequence.

## Phylogenetic relationships

The phylogenetic relationships among the species belonging to clade 3 of Aphelenchoididae based on combined Bayesian analysis are shown in Fig 1. The phylogenetic status of *S. shigaensis* n. sp. within clade 3 was basically consistent among three phylogenetic analyses, i.e., combined Bayesian analysis (Fig 1) and separate analyses (Bayesian trees are provided in S5 and S6 Figs), and corresponded well with their sequence homology (above).

*Seinura shigaensis* n. sp., *S. caverna* and *S. persica* formed a well-supported clade (100% posterior probability support), and the new species was close to *S. caverna*. Two newly established strains of *S. caverna* clearly formed a clade with the type strain. *S. italiensis* belonged to an intrageneric clade apart from *S. caverna* clade and was close to the other *S. italiensis* population.

## Taxonomic description

*Seinura shigaensis* Kanzaki, Ekino, Hamaguchi & Takeuchi-Kaneko n. sp. urn:lsid:zoobank. org:act:AAB2E70C-5969-44D7-8A85-92614FF16C0A (Figs 2–5)

**Description.** *Adult*. Medium- to large-sized species, 644–736 and 787–897 μm in length for male and female, respectively. Body cylindrical, ventrally weakly arcuate when killed by heat treatment. Cuticle of moderate thickness for genus, annulated, lateral field with three incisures. Cephalic region distinctly offset, separated by a clear constriction. Lip separated into six (two dorsal, two subventral and two lateral) equal-sized sectors, roundish rectangular to inverted trapezium in lateral view, ca twice as broad as high. Stylet with wide lumen comprising a short cone ca 43% (41.0–45.9 and 38.9–47.5 for male and female, respectively) of total stylet length and a shaft without clear basal swelling, but appearing slightly expanded at base, and not clearly separated from procorpal tube. Stylet lumen clearly opening ventrally with a large (ca 1/3 of cone length) opening. Procorpus cylindrical, with clear procorpal tube at middle of cylinder, ca two stylet lengths (ca two metacorpal lengths) long, ending in a well developed metacorpus (= median bulb). Metacorpus oval with glandular anterior third and muscular posterior two-thirds, and crescent-shaped metacorpal valve slightly posterior to middle at ca 65% of metacarpal length from its anterior end. Dorsal pharyngeal gland orifice

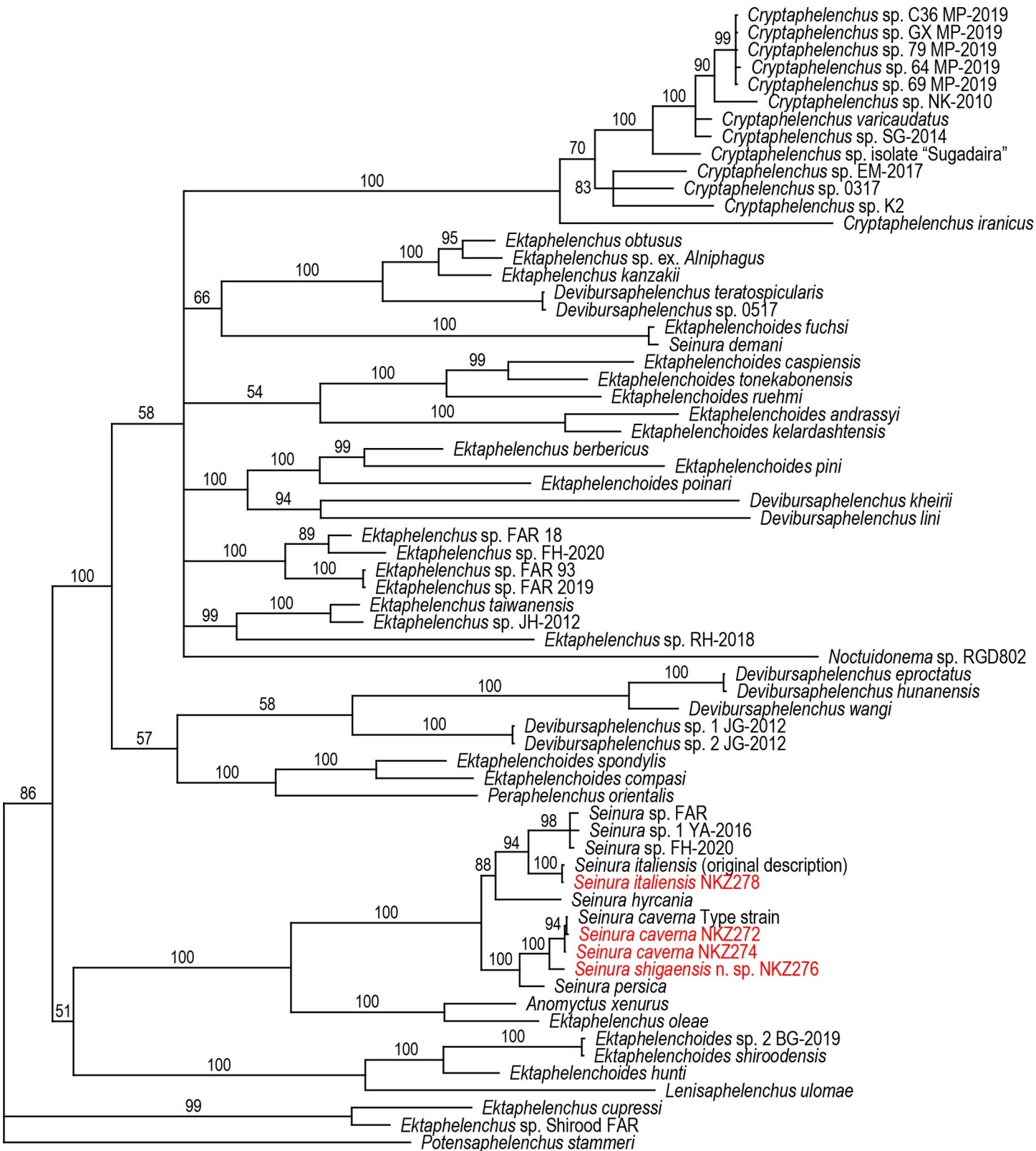

10% difference within *ca* 2.6 kbs of aligned near full length of SSU and D2-D3 LSU

**Fig 1. Combined Bayesian tree inferred from SSU and D2-D3 LSU under GTR+G+I model.** Analytical parameters are as follows: lnL = -15801.98017; freqA = 0.25; freqC = 0.18; freqG = 0.29; freqT = 0.28; R(a) = 0.58; R(b) = 2.75; R(c) = 1.08; R(d) = 0.25; R(e) = 2.58; R(f) = 1.00; Pinva = 0.29; Shape = 0.87 for SSU; and lnL = -11963.5527; freqA = 0.26; freqC = 0.19; freqG = 0.27; freqT = 0.28; R(a) = 0.58; R(b) = 2.33; R(c) = 1.08; R(d) = 0.42; R(e) = 2.92; R(f) = 1.00; Pinva = 0.43; Shape = 0.52 for D2-D3 LSU. Posterior probability values exceeding 50% are given on appropriate clades.

opening into lumen of metacorpus at middle of glandular part of metacorpus or slightly posterior. Pharyngo-intestinal junction immediately posterior to metacorpus. Dorsal pharyngeal gland overlapping intestine dorsally, ca 5–6 metacorpal lengths long. Nerve ring surrounding pharyngeal glands and intestine at ca 0.5 stylet lengths (ca 0.5 metacorpal lengths) posterior to pharyngo-intestinal junction. Hemizonid distinct in permanently mounted material, ca 1 stylet length (ca 1 metacorpal length) posterior to metacorpus. Secretory-excretory pore located ventrally at level of posterior two-thirds (muscular part) of metacorpus.

*Male.* Tail strongly recurved ventrally when killed by heat. Gonad on right of intestine, single, outstretched. Body strongly ventrally arcuate in tail region. Gonad single, on the right of intestine, composed by testis and *vas deferens* from anterior. Anterior end of testis outstretched (8 out of 10 type specimens) or reflexed (2 out of 10), Spermatocytes arranged in single row in anterior 2/3 of testis, then developed sperm tightly packed in multiple rows in the later part of the testis. *Vas deferens* occupying ca 1/4 of gonad, composed by relatively large and flattened cells, often harbouring sperm. The posterior end of *vas deferens* fused with the posterior end of intestine to form a short and simple cloacal tube. Spicules mitten-shaped in lateral view, paired, separate. Condylus broad, rounded, straight from dorsal contour of dorsal limb (lamina), i.e., no dorsal truncation observed. Rostrum triangular with blunt tip directed ventrally. Condylus and rostrum forming a well developed capitulum with clear depression in middle. Blade of spicule (calomus-lamina complex) smoothly ventrally arcuate, smoothly tapering to narrowly rounded (blunt) distal tip. Gubernaculum or apophysis absent. Tail ventrally arcuate, smoothly tapering in anterior 1/3, and distal 1/3 narrowing to form a spike-like projection. Bursal flap absent. Seven (a ventral single papilla and three pairs) conspicuous genital papillae present, a ventral single papilla (P1) at a little less than cloacal body diam. Anterior to cloacal slit; first subventral pair (P2) located at level of cloacal slit (adcloacal), second subventral pair (P3) located *ca* mid way of tail length from cloacal slit, third ventral pair (P4) midway between P3 and root of tail spike. All papillae papilliform, P4 slightly smaller than other two pairs.

*Female.* Reproductive tract located to right of intestine, comprising ovary, oviduct, spermatheca, crustaformeria, uterus, vagina + vulva and post-uterine sac from anterior. Ovary single, anteriorly outstretched (9 out of 10 type specimens) or anteriorly reflexed (1 out of 10). Oocytes present in single row in whole parts of ovary, but arranged alternately dorsally and ventrally in anterior part of ovary and seemingly arranged as two rows. Oviduct tube-like, connecting ovary and spermatheca, sometimes occupied by well developed oocytes. Spermatheca formed by distinctive thick tissue, forming an expansion on right of gonad, i.e., a clearly closed anterior end can be observed on right-hand side of gonad, but not forming a clear branch, slightly irregular oval in shape, sometimes filled with well developed sperm. Crustaformeria not conspicuous, formed by relatively large and rounded cells, sometimes containing sperm. Uterus with thick wall. Vagina slightly inclined anteriorly, junction of uterus, post-uterine sac and vagina usually closed with no special structure such as a pair of three-celled structures. Vulva a simple slit in ventral view, lacking any flap apparatus in lateral or ventral views. Post-uterine sac varying in length among individuals, 2.9–3.5 vulval body diam. long, extending for ca half (46–58%) of vulva to anus distance, sometimes filled with sperm. Anus small, dome-shaped slit in ventral view. Tail ca 7.5–9.1 anal body diam. long, straight or weakly recurved ventrally when killed by heat, cylindrical, forming elongate conoid, smoothly tapering section to filiform tip.

**Morphometrics.** Morphometric values were summarized in Table 1.

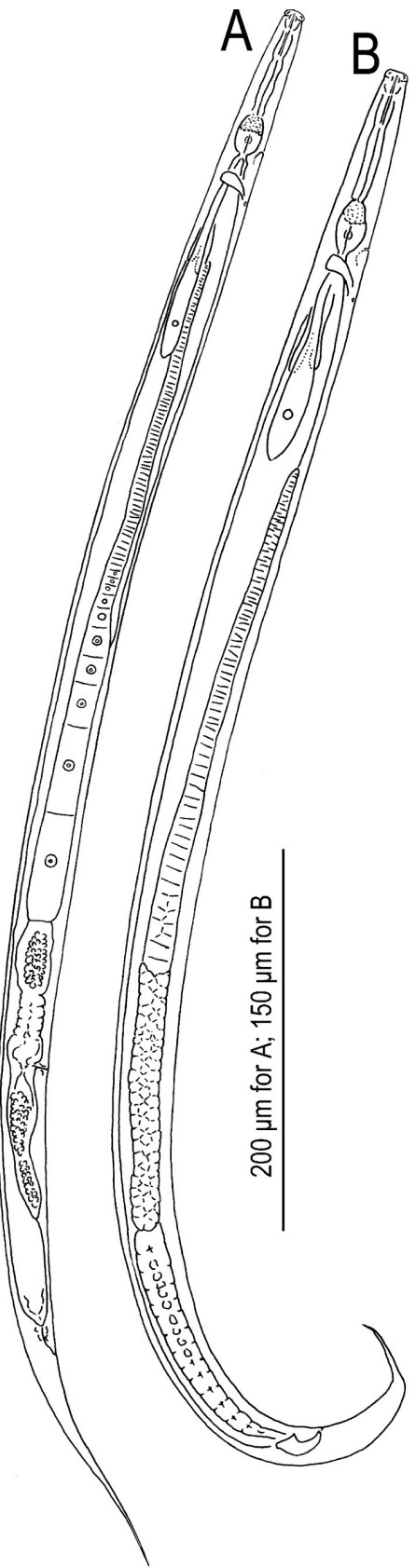

**Fig 2. Right lateral view of adults of Seinura shigaensis n. sp.** A: Entire body of female; B: Entire body of male.

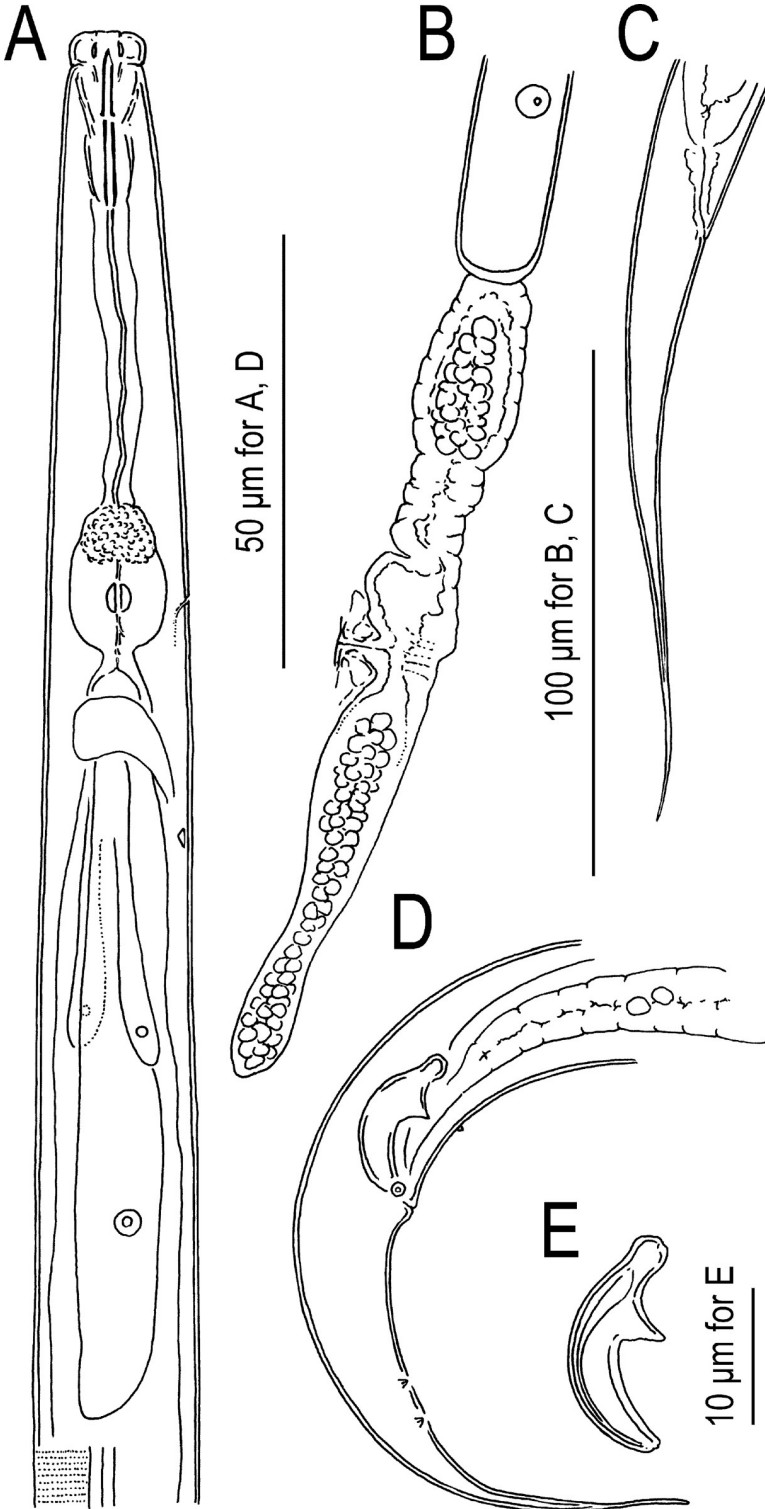

**Fig 3. Adults of Seinura shigaensis n. sp.** A: Anterior region in right lateral view; B: Female vulval region in left lateral view; C: Female tail in right lateral view; D: Male tail in right lateral view; E: Male spicule in right lateral view.

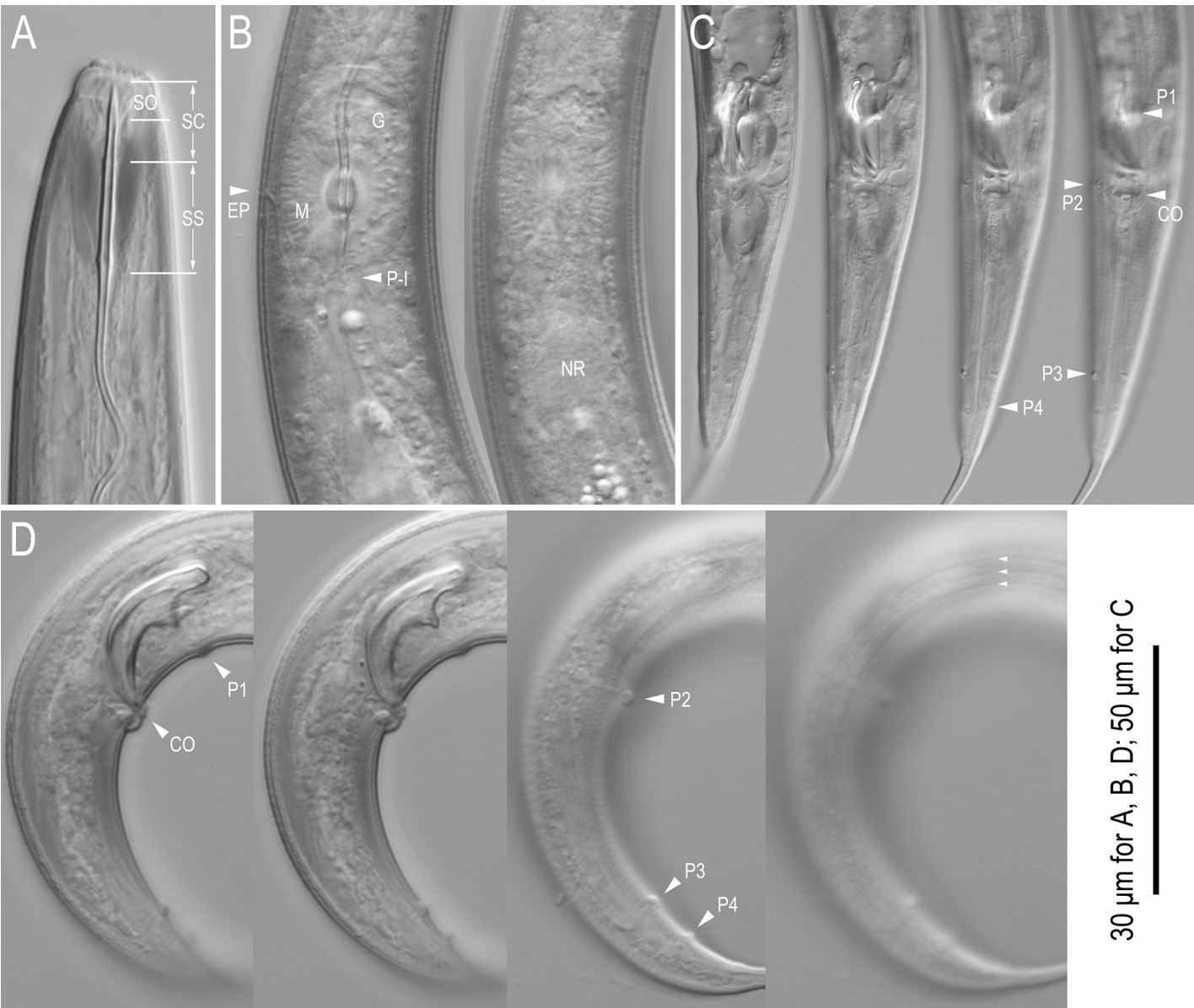

**Fig 4. Live adults of *Seinura shigaensis* n. sp.** A: Lip and stylet region in right lateral view; B: Metacorpal region in left lateral view in two different focal planes; C: Male tail in right lateral view in four different focal planes. Abbreviations are as follows: SO = stylet opening; SC = stylet conus; SS = stylet shaft; EP = secretory-excretory pore; G: glandular part of median bulb; M = muscular part of median bulb; P-I = pharyngo-intestinal junction; NR = nerve ring; CO = cloacal opening; P + number = genital papillae.

**Type material.** Holotype male, four paratype males and five paratype females deposited in the USDA Nematode Collection, Beltsville, MD, USA, with USDANC collection numbers *Seinura shigaensis* T-744t (holotype male), T-7430p-7433p (four paratype males) and T-7434p-7438p (five paratype females). Five paratype males and five paratype females deposited in the Forest Pathology Laboratory Collection, FFPRI, Tsukuba, Japan, with collection numbers *Seinura shigaensis* M01-05 (paratype males) and F01-05 (paratype females).

**Voucher material.** In addition to type materials, several mounted and unmounted specimens of males and females and cultured materials are available from the Kansai Research Center, FFPRI (N. Kanzaki) upon request.

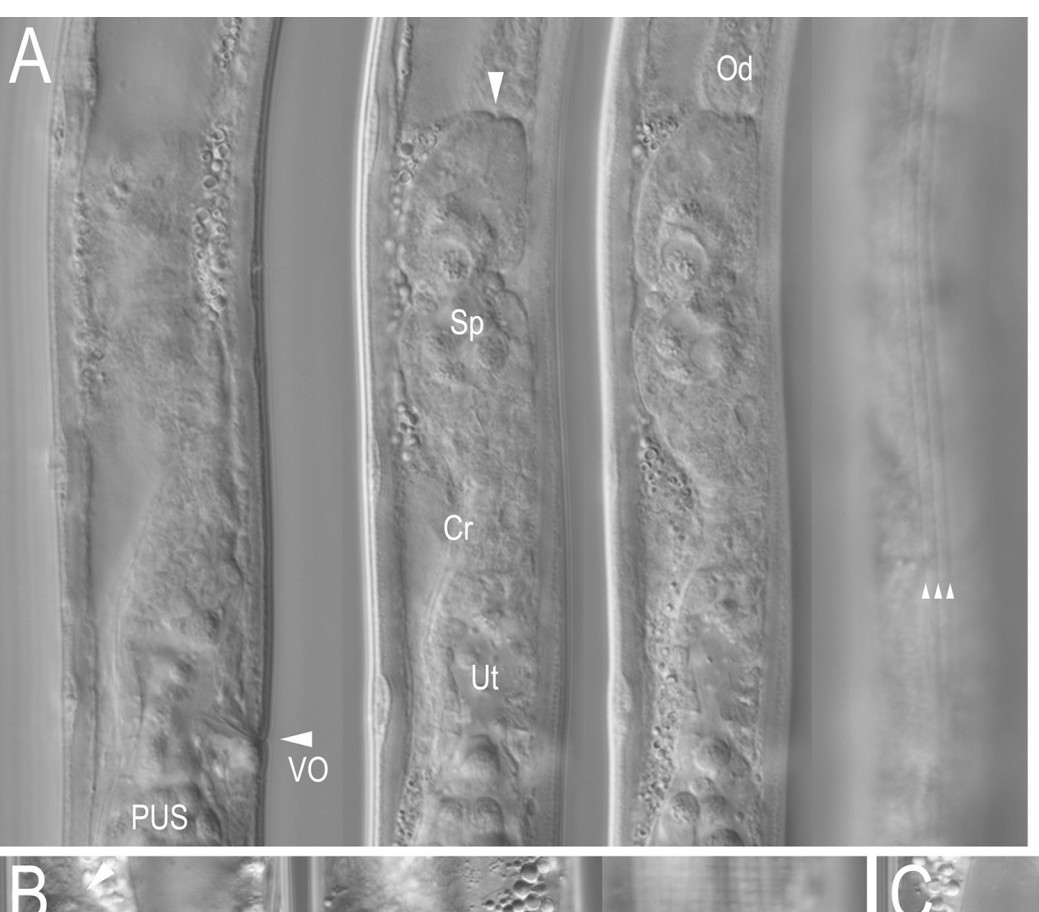

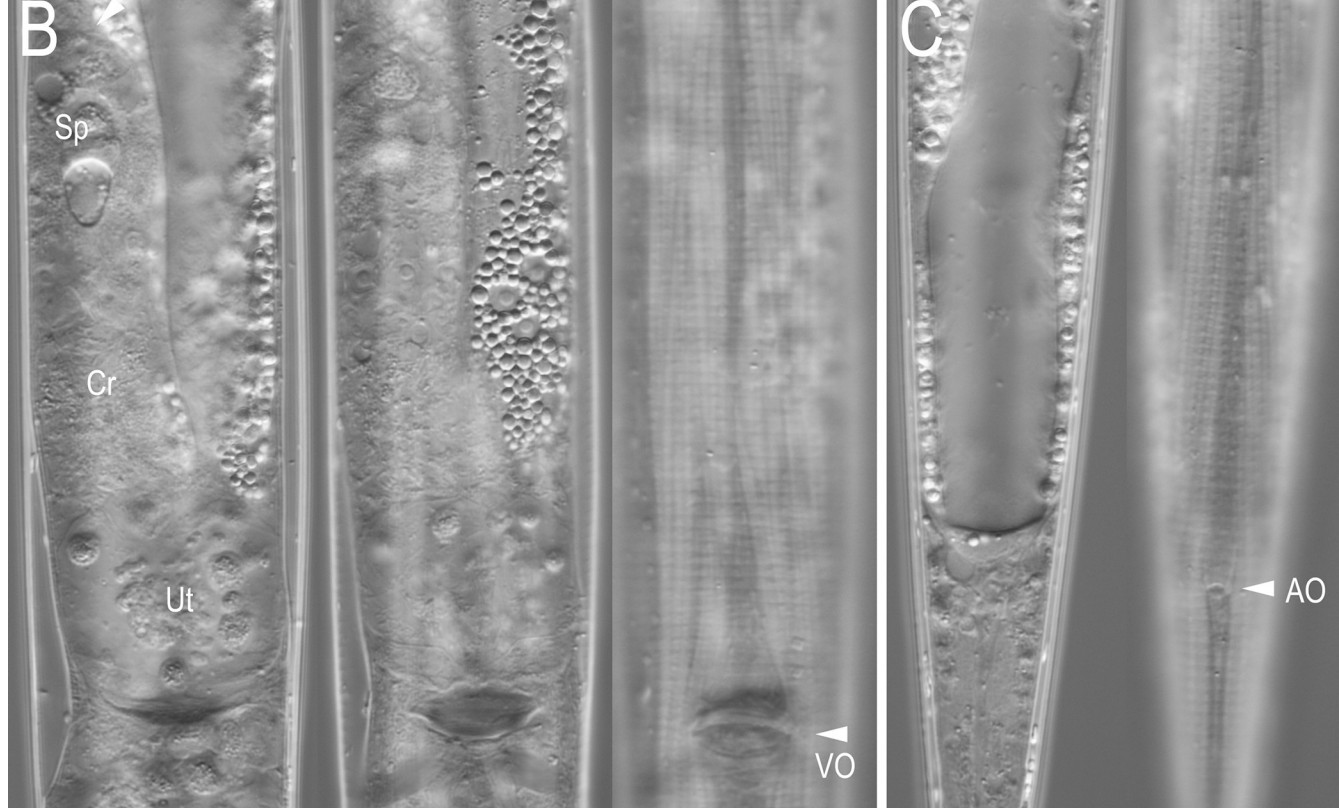

**Fig 5. Live adult female of *Seinura shigaensis* n. sp.** A: Right lateral view of vulval region in four different focal planes (anterior end of spermatheca is suggested with and arrowhead); B: Ventral view of vulval region in three different focal planes; C: Ventral view of anal region in two different focal planes (Lateral lines are suggested with arrowheads). (Abbreviations: VO = vulval opening; PUS = post-uterine sac; Sp = spermatheca; Cr = crustaformeria; Od = oviduct.

**Type habitat and locality.** The type material was obtained from a laboratory population of *S. shigaensis* n. sp. strain NKZ276. The population was established from more than 50 individuals collected from organic soil containing decomposed plant material, which was collected from a hollow on the trunk of *Quercus serrata* Murray from Makino, Shiga (GPS coordinates: 35˚28'41" N, 136˚01'48" E; altitude: 156 m a.s.l.), Japan on 8, March, 2019.

**Table 1. Morphometrics of type specimens of *Seinura shigaensis* n. sp.**

| Character | Male | | Female |
|---|---|---|---|
| | Holotype | Paratypes | Paratypes |
| n | - | 9 | 10 |
| L | 711 | 701 ± 34 (644–736) | 842 ± 28 (787–897) |
| a | 38.3 | 35.6 ± 2.3 (30.9–38.4) | 31.5 ± 1.2 (29.1–33.3) |
| b | 9.6 | 9.7 ± 0.3 (9.2–10.2) | 10.5 ± 0.4 (10.1–11.3) |
| c | 10.3 | 10.5 ± 0.5 (9.8–11.1) | 7.3 ± 0.3 (6.9–7.9) |
| c' | 4.8 | 4.9 ± 0.2 (4.7–5.1) | 8.4 ± 0.5 (7.5–9.1) |
| T or V | 62.9 | 65.9 ± 2.9 (60.6–71.4) | 68.7 ± 0.4 (68.0–69.2) |
| M | 43.5 | 44.5 ± 1.4 (41.0–45.9) | 42.8 ± 2.5 (38.9–47.5) |
| Maximum body diam. | 18.5 | 19.8 ± 1.4 (16.8–21.3) | 26.7 ± 1.2 (24.1–29.0) |
| Lip diam. | 8 | 8.2 ± 0.2 (7.7–8.4) | 9.6 ± 0.2 (9.1–9.8) |
| Lip height | 4.5 | 4.1 ± 0.3 (3.5–4.5) | 4.0 ± 0.3 (3.5–4.5) |
| Lip height/diam. | 1.8 | 2.0 ± 0.2 (1.8–2.3) | 2.4 ± 0.2 (2.1–2.7) |
| Stylet conus | 9.4 | 9.4 ± 0.4 (8.7–9.8) | 10.3 ± 0.3 (9.8–10.8) |
| Stylet length | 21.7 | 21.0 ± 0.7 (19.6–21.7) | 24.2 ± 1.4 (21.3–25.9) |
| Metacorpus diam. | 11.2 | 11.2 ± 0.5 (10.5–11.9) | 13.6 ± 0.4 (12.9–14.3) |
| Metacorpus length | 17.5 | 17.3 ± 0.8 (16.1–18.2) | 20.5 ± 2.2 (14.7–22.4) |
| Metacorpus length/diam. ratio | 1.6 | 1.6 ± 0.1 (1.4–1.7) | 1.5 ± 0.2 (1.1–1.7) |
| Secretory-Excretory pore from anterior end | 61 | 64 ± 3.6 (59–69) | 71 ± 3.7 (66–78) |
| Nerve ring from anterior end | 81 | 81 ± 2.7 (76–85) | 91 ± 1.5 (87–93) |
| Hemizonid from anterior end | 96 | 95 ± 4.3 (86–101) | 104 ± 3.8 (97–111) |
| Testis or ovary length | 447 | 463 ± 38 (390–523) | 449 ± 26 (406–480) |
| Reflexed part of gonad | 0 | 32, 47 (n = 2) | 20 (n = 1) |
| Cloacal or anal body diam. | 14.3 | 13.6 ± 0.4 (12.9–14.3) | 13.8 ± 0.4 (13.3–14.7) |
| Tail length | 69 | 67 ± 2.0 (64–70) | 116 ± 3.8 (110–120) |
| *Vas deferens* length | 123 | 121 ± 14 (85–139) | - |
| % of *vas deference* to total gonad | 27.5 | 26.0 ± 1.9 (21.9–28.1) | - |
| Spicule (chord) | 18.9 | 18.2 ± 0.7 (16.8–18.9) | - |
| Spicule (curved along median line) | 18.2 | 17.6 ± 0.7 (16.1–18.5) | - |
| Vulval body diam. | - | - | 24.8 ± 1.4 (22.0–27.6) |
| Vulva-anus distance | - | - | 148 ± 7.5 (137–164) |
| Post-uterine sac (PUS) length | - | - | 80 ± 6.6 (63–87) |
| PUS % to vulva-anus distance | - | - | 54 ± 3.5 (46–58) |
| PUS / vulval body diam. | - | - | 3.2 ± 0.2 (2.9–3.5) |

All measurements are in μm and in the form: mean ± s.d. (range).

**Diagnosis and relationships.** *Seinura shigaensis* n. sp. is characterised by a lack of a clear swelling at the base of the stylet, position of the secretory-excretory pore (located at the level of the posterior two-thirds [muscular part] of the metacorpus), long post-uterine sac in the female, female tail elongated conoid with a filiform tip, male spicule with relatively large and broad condylus, blunt and triangular rostrum, and smoothly ventrally curved and a wide blade (calomus/lamina complex), number and arrangement of male genital papillae, i.e., three paired papillae present at adcloacal (P2), mid-tail (P3) and between P3 and root of tail spike(P4), and spike-like extension of the distal part of the male tail. In addition to these typological characters, the new species is gonochoristic.

*Seinura* currently contains ca 50 nominal species, some of which are typologically very similar to each other [3, 21, 22]. In addition, some species have been described based only on female characters, as the male has not been identified. In addition, male tail characters are similar to each other in the genus, or not sufficiently described except for several characters, e.g., the number and arrangement of genital papillae. Therefore, we initially used female (hermaphroditic or possibly parthenogenetic) characters for comparison, according to the character list provided in previous studies [3, 10].

Females of the new species have a combination of the following characters: *i*) three-lined lateral field, *ii*) secretory-excretory pore located at the level of the posterior two-thirds of the metacorpus, *iii*) long post-uterine sac occupying approximately half of the vulva-anus distance, and *iv*) tail elongated conoid with a filiform tip. The female of the new species shares these characters with *S. caverna*; *S. chertkovi* Dmitrenko; *S. christiei* J.B. Goodey; *S. italiensis*; *S. steineri* Hechler in Hechler & Taylor; and *S. tenuicaudata* (de Man) J.B. Goodey [3, 10, 23–28].

In the following typological and morphometric comparisons, morphometric values are provided in a form "average (range)", and only average or range is given in the original descriptions of several species.

Typologically and phylogenetically, *S. shigaensis* n. sp. is almost a cryptic species of *S. caverna*, and can be distinguished from *S. caverna* only by its female b value, 10.5 (10.1–11.3) vs 9.1 (8.5–9.9) and reproductive mode, which is gonochoristic vs hermaphroditic [3]. Although only one male has been reported for *S. caverna*, the new species is distinguished from the species by the male b value, 9.7 (9.2–10.2) vs 8.2, male c' value, 4.9 (4.7–5.1) vs 4.5, and the presence *vs*. absence of P1 genital papilla, although the last character should be confirmed in *S. caverna* because of the material condition [3]. In addition, the new species is distinguished from *S. caverna* based on its phylogenetic status (Fig 1, S1 and S2 Figs), i.e., clear differences in the molecular sequences of the SSU and D2-D3 LSU tentative barcoding regions.

The new species is distinguished from *S. chertkovi* by a larger body length, 842 (787–897) vs 615–700 μm in females and 701 (644–736) vs 489–550 μm in males, female a value, 31.5 (29.1–33.3) vs 35.0–41.0, female b value, 10.5 (10.1–11.3) vs 8.2–9.0, female c value, 7.3 (6.9–7.9) vs 8.2–10.0, male c' value, 4.9 (4.7–5.1) vs 3.2, male spicule larger in chord 18.2 (16.8–18.9) vs ca 10.0 μm, female V value, 68.7 (68.0–69.2) vs 72.9–73.0 and the arrangement of male genital papillae, 1+2+2+2 vs 2+2+2+2 [23].

The new species is distinguished from *S. christiei* by its slightly smaller body length 842 (787–897) vs 930–1,000 μm, male b value 9.7 (9.2–10.2) vs 7.8–8.7, female and male c values, 7.3 (6.9–7.9) vs 9.0–10.0 in females and 10.5 (9.8–11.1) vs 12.5–15.5 in males, female and male c' values, 8.4 (7.5–9.1) vs 7.0 in females and 4.9 (4.7–7.5) vs 4.2 in males, and a longer female stylet 24.2 (21.3–25.9) vs 16.0–19.0 μm in females and 21.0 (19.6–21.7) vs 14.0–16.0 μm in males [24, 25].

The morphometric values of Italian and Japanese populations of *S. italiensis* are different from each other (S2 Table), and these populations are compared separately with *S. shigaensis*

n. sp. The new species is distinguished from Italian population of *S. italiensis* by its larger female and male body lengths, 842 (787–897) vs 522 (469–590) μm in females and 701 (644–736) vs 477 (407–565) μm in males, female and male b values, 10.5 (10.1–11.3) vs 7.0 (6.3–7.5) in females and 9.7 (9.2–10.2) vs 6.7 (5.9–7.6) in males, female and male c' values, 8.4 (7.5–9.1) vs 5.6 (3.9–6.5) in females and 4.9 (4.7–5.1) vs 3.6 (3.1–4.2) in males, slightly smaller female V values, 68.7 (68.0–69.2) vs 72.2 (69.7–57.3), longer female tail 116 (110–120) vs 58.3 (43.6–72.0) μm, and larger male spicule, 17.6 (16.1–18.5) vs 14.5 (12.7–15.8) μm in curve and 18.2 (16.8–18.9) vs 14.1 (12.6–15.0) μm in chord [10]. The new species is also distinguished from the Japanese population of *S. italiensis* by its male c value, 10.5 (9.8–11.1) vs 11.8 (11.1–12.7), slightly shorter stylet, 24.2 (21.3–25.9) vs 27.4 (26.0–30.6) μm in females and 21.0 (19.6–21.7) vs 23.5 (22.5–26.5) μm in males and 4.9 (4.7–5.1) vs 3.6 (3.1–4.2) in males, slightly smaller metacorpus of males, diam. x length = 11.2 (10.5–11.9) x 17.3 (16.1–18.2) vs 11.6 (11.2–12.2) x 20.3 (19.4–22.4) μm, and slightly longer male tail, 67 (64–70) vs 58 (53–60) μm. In addition, the new species is distinguished from *S. italiensis* based on its phylogenetic status (Fig 1, S1 and S2 Figs), i.e., a clear difference in the molecular sequences of the SSU and D2-D3 LSU tentative barcoding regions.

The originally described and redescribed populations of *S. tenuicaudata* show a slight difference in body length, i.e., both sexes are slightly larger in the original description [26, 27]. However, *S. shigaensis* n. sp. is distinguished from both populations of *S. tenuicaudata* by its slightly smaller female V value, 68.7 (68.0–69.2) vs 72.0 for the original description and 70.0–78.0 for the redescription) and male c' value, 4.9 (4.7–5.1) vs 4.1 and 3.9 for the original description and redescription, respectively [26, 27].

The new species is typologically identical to *S. steineri*, i.e., the above key characters and key morphometric values overlap. However, *S. shigaensis* n. sp. is distinguished from the species by its gonochoristic *vs* androdioecious reproductive mode [25].

## Discussion

Four strains of *Seinura* spp. were isolated in Makino, Shiga from nutritional soil (*S. shigaensis* n. sp. NKZ276) of an insect carcass bait applied to a soil sample from a *Q. variabilis* stand (*S. caverna* NKZ274) and from decomposed woody material (*S. caverna* NKZ272 and *S. italiensis* NKZ278).

*Seinura italiensis* was originally described from China and found in medium soil of *Olea europaea* L. exported from Italy [10]. In contrast to the packing materials often left in open spaces and easily contaminated by small organisms, medium soil is difficult to contaminate at the entry port, and thus, the nematode is likely native to its originating country, Italy. The worldwide geographical distribution of the nematode has not been clarified sufficiently because of a lack of sampling. However, in the case of the diplogastrid genus *Pristionchus* Kreis, androdioecious species are generally distributed widely, and the distribution of gonochorists is relatively limited to the continent or subcontinent level [29]. Gonochorists in the rhabditid genus *Caenorhabditis* Osche, occasionally have quite a wide distribution [30].

In the present study, *S. italiensis* was found in rotten wood on the institute campus in Kyoto, *ca* 40 km away from the nearest port or international trade airport. As it is difficult for an introduced species to reach an isolated location, two possibilities are suggested for its geographical origin. First, the nematode has a wide distribution range, native to Japan and Italy. Second, the species was introduced (either from Japan to Italy or Italy to Japan) via international trade and disseminated via domestic trade. In fact, some microbe-feeding nematodes have been found in commercially distributed materials in Japan [31]. In addition, the genetic variation between Italian and Japanese populations of *S. italiensis* is smaller than that among domestic strains of *S. caverna*.

In contrast to the above two bacteriophagous genera and other clade 3 aphelenchoidids, which are often found in arthropod carriers, almost no information has been provided for the carrier interaction in *Seinura* spp. [21, 32–35]. Therefore, the distribution mode of the genus remains unknown. Further field collections and biological analyses, particularly on the transmission and distribution modes of *Seinura* spp. are necessary to clarify their distribution and biological interactions in the natural environment. Typological characters of the Japanese population of *S. italiensis* fit well to the original description (S1–S4 Figs) [10]. However, its morphometric values are different from the original description (S2 Table) [10]. The materials for the original description were obtained from the substrate, medium soil exported from Italia to China [10], and the materials for the present study are laboratory strains collected from a culture. Thus, the difference, generally larger in the present study, is possibly because of the nutritional condition, i.e., the laboratory population is well-fed, and grown larger. This also suggests that the body size and related morphometric values vary due to condition, and the values are necessary to be used carefully, considering the material condition.

Two strains of *S. caverna* (NKZ272 and NKZ274) were found at the same locality, the institute campus, but these two strains exhibited some minor molecular differences. The SSU sequence of *S. caverna* NKZ272 was identical to that of the type strain, but that of *S. caverna* NKZ274 showed a 1 bp difference from the sequences of the other two. In addition, the mtCOI gene sequences showed a similar tendency, i.e., that of NKZ274 was slightly different from those of the other two. However, considering the similarity of the D2-D3 LSU sequences (1 bp difference), these strains are likely conspecifics, although a hybridisation experiment was not applicable because no males were found in either of these two strains during observations.

Intraspecific variation in the SSU sequence is rather rare, and even two different species may have identical SSU sequences [36]. Interestingly, the variation was found between two populations collected from the same locality (ca 50 m apart). Considering the rareness of males [3], this variation suggests that these strains (populations) would be difficult to outcross with each other, and that natural and independent mutations are accumulating within the population (or a line).

Further collections of local and more global *S. caverna* populations followed by a detailed population genetic analysis will provide information on diversification of the species. In addition, the function of spontaneous males, e.g., fertility, is also necessary to understand the outcross and reproductive ecology of the species.

*Seinura caverna* is the only cultured hermaphroditic aphelenchoidid predator, and it has many interesting biological and physiological characters, e.g., feeding behaviour and digestive system [3]. In addition, some clade 3 aphelenchoidid predators have a predator-specific cuticle structure [37].

The *Seinura shigaensis* n. sp. found in the present study is fairly close phylogenetically to *S. caverna*; therefore, the new species may be a good comparative system for *S. caverna*. In addition, two strains of *S. caverna* showing clear intraspecific variation could also be experimental materials to compare with the type strain. Further biological surveys and the establishment of isogenic lines of these nematode species will provide a satellite model system for analysing predatory species and their evolutionary biology.

## Supporting information

**S1 Fig. Anterior region of Japanese population of *Seinura italiensis*.** A: Lip and stylet region; B: Metacorpal region in three different focal planes. Abbreviations are as follows: SO = stylet opening; SC = stylet conus; SS = stylet shaft; EP = secretory-excretory pore; G: glandular part of median bulb; M = muscular part of median bulb; P-I = pharyngo-intestinal junction;

NR = nerve ring; arrowheads = lateral lines.
(TIF)

**S2 Fig. Male tail of Japanese population of *Seinura italiensis*.** A: Lateral view in three different focal planes; B: Ventral view in four different focal planes. Abbreviations are as follows: CO = cloacal opening; P + number = genital papillae.
(TIF)

**S3 Fig. Female characters of Japanese population of *Seinura italiensis*.** A: Ventral view of vulval region in three different focal planes; B: Lateral view of vulval region in two different focal planes; C: Ventral view of rectal-anal region in two different focal planes. Abbreviations are as follows: Sp = spermatheca where anterior end is indicated with an arrowhead; M: vulval muscle; VO = vulval opening; AO = anal opening; Rec = rectum.
(TIF)

**S4 Fig. Typological characters of Japanese population of *Seinura italiensis*.** A: Entire body of female; B: Entire body of male; C: Stylet; D: Anterior region; E: Female vulval region; F: Female tail; G: Male tail; H: Male spicule; I: Male tail in ventral view. All subfigures except for H are in right lateral view.
(TIF)

**S5 Fig. Bayesian tree inferred from near-full-length SSU.** The substitution model and analytical parameters are same as the combined tree (Fig 1). Posterior probability support exceeding 50% are presented on appropriate clades.
(TIF)

**S6 Fig. Bayesian tree inferred from D2-D3 LSU.** The substitution model and analytical parameters are same as the combined tree (Fig 1). Posterior probability support exceeding 50% are presented on appropriate clades.
(TIF)

**S1 Table. The molecular sequences of clade 3 aphelenchoidid species compared in the present study.**
(DOCX)

**S2 Table. Morphometric values for two populations of *Seinura italiensis*.**
(DOCX)

## Acknowledgments

The authors sincerely thank Ms Yoshiko Shimada, KRS-FFPRI for her technical assistance in culturing the materials and morphometric analysis.

## Author Contributions

**Conceptualization:** Natsumi Kanzaki, Taisuke Ekino, Keiko Hamaguchi, Yuko Takeuchi-Kaneko.

**Data curation:** Natsumi Kanzaki.

**Formal analysis:** Natsumi Kanzaki.

**Funding acquisition:** Natsumi Kanzaki, Yuko Takeuchi-Kaneko.

**Investigation:** Natsumi Kanzaki, Taisuke Ekino, Keiko Hamaguchi.

**Methodology:** Natsumi Kanzaki, Taisuke Ekino, Keiko Hamaguchi.

**Project administration:** Yuko Takeuchi-Kaneko.

**Resources:** Natsumi Kanzaki, Taisuke Ekino, Keiko Hamaguchi.

**Supervision:** Natsumi Kanzaki, Yuko Takeuchi-Kaneko.

**Validation:** Natsumi Kanzaki.

**Visualization:** Natsumi Kanzaki.

**Writing – original draft:** Natsumi Kanzaki.

**Writing – review & editing:** Natsumi Kanzaki, Taisuke Ekino, Keiko Hamaguchi, Yuko Takeuchi-Kaneko.

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
