## [Decision Letter · Decision Letter 0]

24 Nov 2020

PONE-D-20-34242

Three Seinura species isolated from Japan with a description of S. shigaensis n. sp. (Tylenchomorpha: Aphelenchoididae)

PLOS ONE

Dear Dr. Kanzaki

Thank you for submitting your manuscript to PLOS ONE. After careful consideration, we feel that it has merit but does not fully meet PLOS ONE’s publication criteria as it currently stands. Therefore, we invite you to submit a revised version of the manuscript that addresses the points raised during the review process.

We look forward to receiving your revised manuscript.

Kind regards,

Ebrahim Shokoohi

Academic Editor

PLOS ONE

Additional Editor Comments:

Dear Dr Kanzaki

I have received the referees comments and based on the comments your paper needs a minor revision. The comments on your paper attached below. Please address the concerns raised by the referees point by point and resubmit your revision.

Kind regards,

Ebrahim

Journal Requirements:

Reviewers' comments:

Reviewer's Responses to Questions

**Comments to the Author**

1. Is the manuscript technically sound, and do the data support the conclusions?

Reviewer #1: Yes

Reviewer #2: Yes

Reviewer #3: Yes

2. Has the statistical analysis been performed appropriately and rigorously? 

Reviewer #1: Yes

Reviewer #2: Yes

Reviewer #3: Yes

3. Have the authors made all data underlying the findings in their manuscript fully available?

Reviewer #1: Yes

Reviewer #2: Yes

Reviewer #3: Yes

4. Is the manuscript presented in an intelligible fashion and written in standard English?

Reviewer #1: Yes

Reviewer #2: Yes

Reviewer #3: Yes

5. Review Comments to the Author

Reviewer #1: This manuscript describes a new Seinura species based on morphological and molecular evidences, and reproductive mode research. It’s well-written and should be published after minor corrections.

1. P12: “A BLAST homology search for NKZ276 (S. shigaensis n. sp.) suggested that this species was close to S. caverna (LC414971: 1643/1655 bp of identity without a sequence gap)”, maybe this refers to SSU, then how about the ITS and LSU results?

2. The new species is very close to S. caverna, the author used b value to separate them, while b value is not an important character. So if their DNA sequences are different enough to separate them, I suggest we call it a cryptic species.

Reviewer #2: A well written and documented paper contributing to the knowledge of the genus Seinura -a predatory nematode genus with intriguing biology. One new species is described, and new data on two known species were provided.

I have only a few remarks and suggestions.

I propose a small change in the title omitting the word “isolated”.

Fig. 1 – p. 13, line 243 – repetition of one sentence is noticed.

No description of male tail of S. shigaensis in the differential diagnosis is provided and this character is not used to differentiate it from the closely related species; also in the diagnosis (P. 21 – line 365) – the term hermaphrodite is used, although the new species is considered as gonorchistic.

Comparisons between the new species and S. Italiensis is based only on metrics of the type population, I think that such comparison should be made with the Japanese population of this species.

My main concern is S. italiensis – the morphometrics of the Japanese population differs substantially from that of type population. This was not discussed in details. Unfortunately, no illustrations have been provided to confirm that these two populations are close enough in their morphology (body habitus, shape of tail etc.), not only genetically in term of obtained rDNA sequences. There are examples in other nematode groups where the used loci (SSU and D2-D3 LSU) are unable to distinguish different species, e.g. some members of X. americanum group.

I recommend using type population for S. italiensis instead original description in phylogenetic trees.

In S Table 2 data for the type population of X. italiensis Vulval body diam. and PUS / vulval body diam. are presented as X ± XX (XX)

Reviewer #3: This is a well prepared and organized manuscript presenting new and significant information on "Three Seinura species isolated from Japan with a description of S. shigaensis n. sp.

(Tylenchomorpha: Aphelenchoididae)" worthy of publication. The new species appears to be new which is characterized by both morphological and molecular means and is clearly differentiated from its closely related species. The survey yielded four new strains of Seinura spp., including two strains of S. caverna, a strain of S. italiensis, and a strain of an undescribed species. The molecular results are quite promizing for the new species indicating phylogenetically very close to S. caverna, and that it could be a good comparative system for S.caverna as a potential satellite model for the predatory nematode. The line drawings and photomicrographs are of excellent quality precisely showing the structures of particular value and as described in the text.

6. PLOS authors have the option to publish the peer review history of their article (what does this mean?). If published, this will include your full peer review and any attached files.

Reviewer #1: No

Reviewer #2: No

Reviewer #3: **Yes: **Zafar Handoo

---

## [Author Response · Author response to Decision Letter 0]

10 Dec 2020

We uploaded the responses as an attached file "Responses". 

Below is copy-paste from the file. 

=====

Reviewer #1: 

This manuscript describes a new Seinura species based on morphological and molecular evidences, and reproductive mode research. It’s well-written and should be published after minor corrections.

1. P12: “A BLAST homology search for NKZ276 (S. shigaensis n. sp.) suggested that this species was close to S. caverna (LC414971: 1643/1655 bp of identity without a sequence gap)”, maybe this refers to SSU, then how about the ITS and LSU results?

Response: 

The LSU (D2-D3) are mentioned in the text (P. 12; after SSU). We added the similarity in ITS to the subsection “Genotyping and sequence variation” in the section “Results”. The ITS of S. shigaensis was closest to that of S. caverna, but second and third closest species were Cryptaphelenchus spp., and no Seinura spp. came up as the close sequences. Thus, only S. caverna was mentioned here for ITS. 

2. The new species is very close to S. caverna, the author used b value to separate them, while b value is not an important character. So if their DNA sequences are different enough to separate them, I suggest we call it a cryptic species.

Response: 

We added some more explanations to the beginning of the differential diagnosis. 

===

Reviewer #2: 

A well written and documented paper contributing to the knowledge of the genus Seinura - a predatory nematode genus with intriguing biology. One new species is described, and new data on two known species were provided.

I have only a few remarks and suggestions.

I propose a small change in the title omitting the word “isolated”.

Response: 

Corrected. 

Fig. 1 – p. 13, line 243 – repetition of one sentence is noticed.

Response: 

Corrected. That could be because of the error occurred in copy-paste from another file. Thank you for catching the error. 

No description of male tail of S. shigaensis in the differential diagnosis is provided and this character is not used to differentiate it from the closely related species.

Response: 

Male tail characters provided in the later half of “diagnosis”, the first paragraph of “diagnosis and relationships”. But the characters are basically very similar to each other in the genus, and in the present case, all species mentioned in the detailed comparison did not show clear typological difference in male tail character from new species (or the characters were not described clearly for the comparison). Thus, we could not use these typological characters, but only some morphometrics. 

We noted the typological similarities in the diagnosis. 

Also in the diagnosis (P. 21 – line 365) – the term hermaphrodite is used, although the new species is considered as gonorchistic.

Response: 

Those were mistakes; the species is gonochorist. Corrected. 

In addition, the type specimen “Holotype hermaphrodite” was also wrong. This was corrected as well. 

Thank you for catching the errors. 

Comparisons between the new species and S. Italiensis is based only on metrics of the type population, I think that such comparison should be made with the Japanese population of this species.

Response: 

The differential diagnosis from Japanese population was added after that from Italian population. 

My main concern is S. italiensis – the morphometrics of the Japanese population differs substantially from that of type population. This was not discussed in details. Unfortunately, no illustrations have been provided to confirm that these two populations are close enough in their morphology (body habitus, shape of tail etc.), not only genetically in term of obtained rDNA sequences. There are examples in other nematode groups where the used loci (SSU and D2-D3 LSU) are unable to distinguish different species, e.g. some members of X. americanum group.

I recommend using type population for S. italiensis instead original description in phylogenetic trees.

Response: 

The type population of S. italiensis was used for the phylogenetic analyses. At current status, no live strain of the typo population is available, and the further study cannot be conducted. 

We provided the drawings (as new S4 Figure) and some more explanations about the difference between populations of S. italiensis to Discussion. 

In S Table 2 data for the type population of S. italiensis Vulval body diam. and PUS / vulval body diam. are presented as X ± XX (XX)

Response: 

Corrected. Vulval body diam. was not given in the original description, and PUS-VBD ratio was noted in the text, and the value was added to the table. Thank you for catching the error. 

===

Reviewer #3: 

This is a well prepared and organized manuscript presenting new and significant information on "Three Seinura species isolated from Japan with a description of S. shigaensis n. sp. (Tylenchomorpha: Aphelenchoididae)" worthy of publication. 

The new species appears to be new which is characterized by both morphological and molecular means and is clearly differentiated from its closely related species. The survey yielded four new strains of Seinura spp., including two strains of S. caverna, a strain of S. italiensis, and a strain of an undescribed species. The molecular results are quite promizing for the new species indicating phylogenetically very close to S. caverna, and that it could be a good comparative system for S. caverna as a potential satellite model for the predatory nematode. The line drawings and photomicrographs are of excellent quality precisely showing the structures of particular value and as described in the text.

Response: 

Thank you for your positive comments.

---

## [Editor Report · Decision Letter 1]

15 Dec 2020

Three Seinura species from Japan with a description of S. shigaensis n. sp. (Tylenchomorpha: Aphelenchoididae)

PONE-D-20-34242R1

Dear Dr. Kanzaki

We’re pleased to inform you that your manuscript has been judged scientifically suitable for publication and will be formally accepted for publication once it meets all outstanding technical requirements.

Kind regards,

Ebrahim Shokoohi

Academic Editor

PLOS ONE

Additional Editor Comments (optional):

Dear Dr Kanzaki

I am pleased to inform you that your paper has been accepted for publication in PLOS ONE. After considering all concerns raised by the referee, the essential changes have been implemented to the manuscript. Therefore, I have accepted you and your colleagues paper. However, please kindly check all the citation and reference parts to be in accordance with PLOS ONE. Some scientific names e.g., p:24 , line 416 is not italicized.

Kind regards,

Ebrahim

Reviewers' comments:

NO COMMENT

---

## [Editor Report · Acceptance letter]

18 Dec 2020

PONE-D-20-34242R1 

Three *Seinura* species from Japan with a description of *S. shigaensis* n. sp. (Tylenchomorpha: Aphelenchoididae) 

Dear Dr. Kanzaki:

I'm pleased to inform you that your manuscript has been deemed suitable for publication in PLOS ONE. Congratulations! Your manuscript is now with our production department. 

Kind regards, 

on behalf of

Dr. Ebrahim Shokoohi 

Academic Editor

PLOS ONE